# Irrational Use of Medications among Adults with Insomnia: An Observational Study at a Sleep Clinic in Mexico

**DOI:** 10.3390/pharmacy12020056

**Published:** 2024-03-27

**Authors:** Lucelya Carvalho-Silva, Ulises Jiménez-Correa, Rafael Santana-Miranda, Ivo Heyerdahl-Viau, Jonatan Benitez-Morales, Mireya García-Casas, Juan Manuel Martínez-Núñez

**Affiliations:** 1Department of Biological Systems, Universidad Autónoma Metropolitana, Campus Xochimilco, Mexico City 04960, Mexico; lucelyasilva.20212001714@uemasul.edu.br (L.C.-S.); iheyerdahl@correo.xoc.uam.mx (I.H.-V.); 2Clinic of Sleep Disorders, School of Medicine, Universidad Nacional Autónoma de México (UNAM), Mexico City 06720, Mexicorafaelsantanamiranda@comunidad.unam.mx (R.S.-M.); 3Department of Hospital Pharmacy, Instituto Nacional de Enfermedades Respiratorias (INER), Mexico City 14080, Mexico; 4Department of Clinical Pharmacy, Facultad de Estudios Superiores, Campus Zaragoza, Universidad Nacional Autónoma de México (UNAM), Mexico City 09230, Mexico

**Keywords:** insomnia, irrational use of drugs, hypnotics, antidepressants, anticonvulsants, antipsychotics

## Abstract

Background: To determine the patterns of irrational use of medications among a sample of adult patients with insomnia. Methods: We included 89 adult patients diagnosed with chronic insomnia who had consumed medications for this disorder during the 12 months prior to admission to a specialized Sleep Disorders Clinic (SDC) in Mexico City. With a 13-item survey, information was gathered on patterns of medication use and irrational use, considering therapeutic indications, dose, route of administration, and duration of treatment. Results: The participants had taken hypnotics (65%), antidepressants (21%), anticonvulsants (8%), and antipsychotics (6%), and 92% had irrational use of their medication. Irrational use was greatest with benzodiazepines and antipsychotics. There were two main types of irrational use: (1) 47% of participants had consumed a drug unsuitable for their condition, although it was almost always prescribed by a doctor, and (2) 43% had consumed a drug for longer than the maximum time recommended. Conclusion: It is worrisome to find that the irrational use of medications to treat insomnia, especially benzodiazepines and antipsychotics is widespread. Although most participants had acquired their medication by prescription, for many the drug was inappropriate to treat their condition. It should be mandatory that patients with insomnia receive specialized medical attention in primary clinical care.

## 1. Introduction

Insomnia is the difficulty in initiating or maintaining sleep, the presence of premature awakenings, and the sensation of non-restorative sleep despite adequate sleeping conditions, with repercussions on daytime functioning. The current classification considers acute and chronic insomnia. The latter affects 3 or more days of the week and for more than 3 months. Treatment is directed at the underlying causes with the support of cognitive behavioral therapy, sleep hygiene measures and specific pharmacological therapy [1].

Insomnia has a prevalence of 22.1% among adults worldwide [2] and 18.8% among adults over 20 years of age in Mexico (according to surveys of the population) [3]. It is the most common sleep disorder. In the short run, it produces daytime fatigue, irritability, dysphoria, tension, and situational depression. Insomnia also has significant long-term health consequences such as increased risk of depression, mood lowering, irritability, poor memory, fatigue, or even more severe consequences such as increased risk of mortal comorbidities (e.g., hypertension), interference with work performance, sleepiness while driving and, therefore, accidents [4]. Due to its adverse effects, this disorder is associated with traffic accidents and an increase in general morbidity and mortality. Insomnia, especially if not effectively treated, leads to considerably greater use of healthcare resources [5].

Insomnia is a frequent motive for consultations in primary care, where in most cases the problem could theoretically be resolved [6]. However, up to 85% of patients with chronic insomnia are never identified because many sleep disorders are related to other existing psychological or physical conditions. Thus, the disorder is often improperly treated or left untreated, causing it to continue and worsen [7].

Chronic insomnia leads some patients to seek care with multiple clinicians, to misuse prescribed medications, and/or to self-medicate in order to solve their problem of sleep deprivation and daytime discomfort. The irrational use of prescription medication or the decision to self-medicate represents a lack of proper control of therapeutic drugs, which can result in abuse and dependence and/or trigger serious adverse reactions, in some cases resulting in hospitalization or even death [8]. There is little information in the literature on the extent of irrational use of medications taken as a pharmacological treatment for insomnia, or the causes of such irrational use. The aim of the current contribution was to determine the patterns of irrational use of medications among a sample of adult patients with insomnia.

## 2. Materials and Methods

This is an observational and prospective study conducted at the Sleep Disorders Clinic (SDC), a specialized service provided to the entire population in the General Hospital of Mexico.

### 2.1. Participants

The sample consisted of 89 adult patients (>18 years of age) with insomnia who had sought medical attention previously. They were interviewed upon their first visit to the SDC between February 2019 and February 2020. Inclusion criteria for participants in this study included a diagnosis of chronic insomnia and a history of consumption of pharmacological agents to treat insomnia during the 12 months prior to admission to the SDC. Patients with a medical history of psychiatric disorders, cognitive disorders, or addiction to other drugs were excluded from the study. The project was approved by the Research and Ethics Committee of the Metropolitan Autonomous University (ID number: CEI.2020.002). All individuals gave written informed consent to participate.

### 2.2. Measurements and Analysis

The definition of the World Health Organization of the Rational Use of Drugs, established in 1985 and still in force was adopted for the current study: “Patients receive medications appropriate to their clinical needs, in doses that meet their own individual requirements, for an adequate period of time, and at the lowest cost to them and their community” [9]. Any deviation from this definition was considered an irrational use of drugs, whether in relation to the therapeutic indications, dose (quantity and frequency), route of administration, or duration of treatment.

The diagnosis of insomnia was made by the SDC staff members based on the medical history of the patient. The staff are specialists in sleep disorders. The Insomnia Severity Index questionnaire was applied [10], and the criteria for diagnosis were in accordance with the third edition of the International Classification of Sleep Disorders [11].

Upon admission to the SDC, data were collected on the gender, age, level of schooling, work activity, work shift, and marital status of the participants. All medications taken by the patients to treat insomnia during the 12 months prior to admission to the SDC were recorded and the pattern of consumption was examined in detail. The information gathered, based on a 13-item survey was related to different aspects of possible irrational use of drugs: the therapeutic indications, dose (quantity and frequency), route of administration, and duration of treatment. The survey also included questions about the way participants acquired their drugs in order to determine whether they were obtained without a prescription.

The information gathered was compared to that contained in the National Compendium of Health Supplies of Mexico [12], the national and international pharmacotherapeutic guidelines [13,14,15,16,17], and studies published in PubMed. In this manner, an evaluation was made of the scientific basis of taking each drug (even an off-label prescription) for the treatment of insomnia or the comorbidity causing insomnia. Additionally, the dose (quantity and frequency) was compared to the therapeutic range found in guidelines and that reported in the literature, and the route of administration and the maximum treatment time was also compared to guidelines and scientific studies.

The manner in which the medications were acquired was classified as correct if they were prescribed by a doctor and incorrect if not. In the absence of a prescription, an inquiry was made as to whether the drug was given to the patient by a friend or relative, obtained through the informal market, or stolen from a healthcare site (pharmacy, hospital or clinic).

The characteristics of the sample population as well as the medications used rationally and irrationally were analyzed by mean values and percentages. A bivariate logistic regression was performed to ascertain the possible association of patient irrational use with age, gender, level of schooling, and marital status, and the manner of acquiring the medication. Odds ratios (OR) and 95% confidence intervals were calculated for each independent variable. A *p*-value < 0.05 was considered significant. Data analysis was performed on Stata 17.0 (StataCorp, College Station, TX, USA).

## 3. Results

The majority of the participants in the study were women (64%). Among all individuals, most were over 60 years of age (55%), married (62%), and dedicated to caring for the home or without a job (52%). Forty-four percent had a limited level of education (not having completed junior high school, or in some cases not having completed elementary school) (Table 1).

The medications used were hypnotics (65%), antidepressants (21%), anticonvulsants (8%), and antipsychotics (6%). The predominant consumption was as follows: clonazepam, alprazolam, and zolpidem among hypnotics (Figure 1A); escitalopram, mirtazapine, sertraline, citalopram, and duloxetine among antidepressants (Figure 1B) and pregabalin and carbamazepine among anticonvulsants (Figure 1C). The most frequently consumed antipsychotic was quetiapina (Figure 1D).

In accordance with professional guidelines and the scientific literature, a surprising 92% of the study participants had irrational use of drugs to treat insomnia during the 12 months prior to consulting the SDC. There were two very common types of irrational use: (1) 47% consumed a drug that was not suitable for their condition (even though they almost always received their medication through a prescription by a doctor), and (2) 43% consumed the drug for a period longer than the maximum recommended treatment time (Figure 2A). Benzodiazepines, zolpidem, and antipsychotics were the drugs with the highest degree of irrational use, while antidepressants and anticonvulsants were more frequently utilized correctly (Figure 2B).

Although 92% of the patients had irrational use of their medications, they almost always received them through a prescription by a doctor. Only a small proportion of participants acquired their medicines through incorrect channels (e.g., as a gift or in unauthorized places (Figure 3)).

The odds ratios were calculated to determine the relation of irrational use of drugs to sociodemographic characteristics and the manner of acquiring medications (Table 2). No significant association was found.

## 4. Discussion

The findings show that the frequency of the irrational use of therapeutic drugs was high among patients with insomnia participating in the present study. Since their former treatment regimen had failed, they sought medical attention at the SDC in Mexico City. Ninety-two percent of study participants had irrational use of their medications for insomnia. Irrational use was most often found in relation to the consumption of a drug not suitable for the patient’s condition or for a period longer than the maximum recommended treatment time. To a lesser extent, an inappropriate dose (quantity and/or frequency) was taken.

Since irrational use was so widespread, no significant associations could be found with the age, gender, level of schooling, or marital status of the participants, or the manner of acquiring the medication.

Despite the fact that limited evidence exists in the literature for the use of benzodiazepines to treat insomnia [18,19] and that there is better scientific evidence supporting the effectiveness of therapeutic alternatives, the participants of the present study had taken benzodiazepines in a higher proportion than any other type of medication. Indeed, benzodiazepines are among the most commonly prescribed drugs worldwide [20]. However, their chronic consumption is a risk factor for accidents and dependency, which has led to the formulation of guidelines cautioning against their prescription for senior citizens [21]. The prescription of benzodiazepines for insomnia is actually an off-label use of the drug, possibly based more on the beliefs and experiences of general practitioners than on scientific evidence.

Two benzodiazepines, clonazepam and alprazolam were taken by a high proportion of the current participants. They are included in the guidelines for clinical practice in Mexico for the management of insomnia. Nevertheless, the same guidelines point out the low quality of scientific evidence in support of the administration of alprazolam, while the evidence is unspecified for clonazepam [15]. Interestingly, the guidelines for clinical practice in Spain indicate that alprazolam has one of the worst risk-benefit ratios in the treatment of insomnia, while clonazepam is not even mentioned [14]. The guidelines of Europe and the U.S.A. do not consider alprazolam or clonazepam as therapeutic options for managing insomnia [16,17].

Of the individuals taking benzodiazepines, 87% were involved in irrational use, mainly because the drug was not suitable for the particular condition of the patient (despite being prescribed by a doctor in almost all cases) or was consumed during an excessive period of time. Various studies have demonstrated the short-term effectiveness of taking benzodiazepines to treat insomnia, for a maximum period of 2–4 weeks [17,22,23]. Although evidence of their long-term efficacy is not supported by the literature, and the risk of physical dependence and addiction is well documented [24,25], the majority of the participants of the present study who took these medications had surpassed the maximum recommended treatment time. Similarly, a previous report pointed out the tendency of doctors to over-prescribe benzodiazepines and the common tendency of patients to utilize them for a period longer than the recommended treatment time. These drugs are often prescribed for insomnia before applying other first- and second-line therapeutic techniques having better evidence of effectiveness in the literature [18]. The evidence is very clear for the effectiveness of cognitive behavioral therapy, relaxation techniques, education about good sleeping habits, and serotonergic agents [16]. Zolpidem is also reported to have definite advantages over benzodiazepines. In fact, the latter is only recommended for treating insomnia when it is severe or proves to be resistant to multiple treatments [18].

Furthermore, the consumption of benzodiazepines has been associated with rebound insomnia, the development of tolerance to the medication, and deregulation of normal sleep architecture. Such adverse effects are reported to occur for months after discontinuing consumption. Even the short-term use of benzodiazepines can decrease sleep time, decrease deep-stage/slow-wave sleep, increase rapid eye movement (REM) sleep latency, increase stage 2 non-REM sleep, and decrease delta count [18].

There is evidence in the literature of various significant complications in people over 60 years of age that result from the use of benzodiazepines. Nevertheless, the majority of participants in the present study who had consumed these drugs were over 60 years old. Complications include dependence as well as the risk of falls and bone fractures related to cognitive deficits (particularly in memory, learning, attention span, and visual-spatial intelligence) and dementia. Moreover, there is a considerable rise in mortality due to all the aforementioned causes [25].

Non-benzodiazepine hypnotics, commonly called Z drugs, represent alternatives to benzodiazepines. They have been investigated for the treatment of insomnia in order to seek options capable of promoting sleep without generating the same adverse effects as benzodiazepines [26]. One Z drug is zolpidem, which was utilized by 12% of the present participants who took hypnotics. Zolpidem is reported to have considerable advantages over benzodiazepines, such as a better profile of adverse reactions, including fewer residual next-day effects (e.g., drowsiness, lethargy, and fatigue) as well as other effects with prolonged use (e.g., rebound insomnia, drug tolerance, and dependency) [26].

Of the participants in the current study who had taken zolpidem, over 60% had irrational use of the drug by taking it for periods longer than the recommended time. It is inadvisable for patients to take zolpidem for over 90 days because the drug is known to trigger delirious behavior, fractures and falls, longer latency to sleep, and shorter duration of sleep [27,28].

Antidepressants were also taken by a high proportion (21%) of participants, though in a lower proportion of the sample than those consuming hypnotics (65%). They are commonly prescribed for insomnia as off-label drugs. Antidepressants are not indicated for insomnia and the evidence of their benefit is uncertain [29]. The prescription of antidepressants for the short-term treatment of insomnia can be beneficial in the presence of symptoms of concomitant depression or a background of depression. Indeed, insomnia may be a prodromal symptom of an episode of depression [14,15,16,17,29]. In all other cases, non-benzodiazepine hypnotics should be preferred after the application of non-pharmacological interventions.

In this study, 35.5% of participants who took antidepressants had irrational use of them. They either utilized them for periods longer than the recommended time or did not show a clinical history or symptoms of depression. The majority of antidepressants consumed were selective serotonin reuptake inhibitors (SSRIs). A systematic review of the Cochrane Library produced no convincing reports on the effectiveness of SSRIs in treating insomnia. The existing randomized controlled trials are of low quality [30]. Among tricyclic antidepressants, only amitriptyline and imipramine were utilized by the subjects of the current study. A review of the Cochrane Library did not discover any reports with evidence of their effectiveness in treating insomnia [30].

In a lesser proportion (8%), anticonvulsants (e.g., carbamazepine, lamotrigine, phenytoin, pregabalin, and gabapentin) were utilized by individuals in the present sample population. According to the National Compendium of Medical Supplies (Compendio Nacional de Insumos para la Salud) in Mexico, carbamazepine, lamotrigine, and phenytoin are not approved for treating insomnia, but rather for convulsive crisis and epilepsy [31]. In the guidelines for the clinical management of insomnia in Mexico, carbamazepine is the only one of the three medications included. However, the indication is for the treatment of night terror, not insomnia [15]. Lamotrigine and phenytoin are not contemplated as therapeutic options for the treatment of insomnia in the guidelines of Mexico, the U.S.A., Spain, or Europe [14,15,16,17].

Regarding the effects of carbamazepine on sleep, the evidence is inconsistent and inconclusive [32], making the utilization of this drug for the treatment of insomnia unjustified. On the other hand, lamotrigine and phenytoin are not indicated for the treatment of insomnia and in some patients have actually caused insomnia or sleep alterations [32,33,34].

Two gabapentinoids, pregabalin and gabapentin are classified pharmaceutically as anticonvulsants but have diverse neuromodulating properties that are very effective. Due to their sedative, hypnotic, and anxiolytic properties, they have been utilized in the management of insomnia, particularly insomnia associated with a clinical manifestation of anxiety, chronic pain, fibromyalgia, or sleep-related movement disorders [35]. Nevertheless, a systematic review (conducted in 2021) does not show any evidence from clinical studies to support the off-label use of these medications to treat insomnia. Moreover, the evidence on safety and effectiveness is limited and inconclusive [35]. According to the guidelines of clinical practice in the U.S.A. and Spain [16,17], the administration of gabapentin to treat insomnia represents an insufficient strategy. Furthermore, the irrational use and abuse of pregabalin are reported to be even greater than that of gabapentin [36].

The consumption of gabapentinoids can have considerable negative consequences. Pregabalin produces a strong addiction in the event that patients with insomnia abuse the drug by relying on its soporific properties. Additionally, discontinuing the drug generates withdrawal symptoms and a rebound of insomnia [37].

Fortunately, the consumption of anticonvulsants was limited to 8% of the participants in this study. It was unjustified in a little over 40% of the cases, given the lack of relation of the symptoms to the uses specified in the guidelines.

Regarding antipsychotics, the participants utilized two second-generation drugs, quetiapine and olanzapine. They are approved by the Council of General Health (Consejo de Salubridad General) in Mexico for the treatment of schizophrenia [31]. In medical practice, the off-label administration of these medications at a low dose has become commonplace for the treatment of insomnia [38,39], even though the clinical evidence of their long-term efficacy and safety is insufficient to justify such use [14,16,17,38]. The current results show that antipsychotics were irrationally used at a higher percentage (~90%) than the other three types of drugs employed in the treatment of insomnia.

Quetiapine was the most frequently consumed antipsychotic. One report described a strong tendency for irrational use of this drug among individuals with insomnia and the association of the drug with adverse effects. Quetiapine creates a propensity to gain weight and might induce a desire to seek other drugs [40]. Weight gain is related to snoring, sleep apnea, and drowsiness [41]. Olanzapine has also been associated with irrational use and dependence [42]. The American Diabetes Association, the American Psychiatric Association, the American Association of Clinical Endocrinologists, the North American Association for the Study of Obesity, and the American Geriatric Society have advised against the use of quetiapine to treat insomnia because of adverse consequences (e.g., metabolic and cardiovascular alterations), found even at low doses. Hence, other drugs should be employed unless schizophrenia is a concomitant condition [43].

Overall, almost half of the medications consumed by the participants were not appropriate for treating their particular condition of insomnia. These medications obviously were not effective, given that the lack of resolution of the problem with sleep was the reason that the patients sought specialized help in the SDC. Of the 47% of the current participants who had taken a medication unsuitable for their particular condition, most acquired it through the correct channel (a medical prescription), and most took benzodiazepines or zolpidem. Similar results were obtained in a study in England. According to the surveys made by researchers on the Internet, among the people who irrationally used therapeutic drugs for insomnia, 66.4% took benzodiazepines or zolpidem, of which 55.2% acquired their medications by prescription [44].

At least 28% of the participants of the present study obtained medications by prescriptions from two or more doctors, leading to an accumulation of drugs and thus the risk of combining them or taking an excessively high dose [45]. One report suggests that doctors should recognize their role as a potential source of benzodiazepines destined for misuse [46]. It is important for doctors to understand the motives of their patients in order to determine the correct regimen.

Bad choices of treatment by doctors affected over 40% of the participants of the current study. This is probably a result of a systemic problem in Mexico and other parts of the world: the doctors who diagnose and treat insomnia are usually general practitioners. In many cases, these doctors lack adequate academic training for diagnosing and treating sleep disorders [47,48]. They often do not discuss the problem with the patients, considering insomnia as a secondary symptom of another disease [49]. The limited doctor-patient communication and the absence of follow-up have negative consequences. Using an unsuitable medication for the treatment of insomnia or a suitable one for an excessive period of time can lead to taking unnecessarily high doses, suffering from adverse reactions, and/or continuing to experience the same problem. Each of these outcomes will result in greater problems and sometimes can give rise to hospitalization or even death [50,51]. Although it was much less common, some of the participants in this study used the incorrect dose or route of administration.

On the other hand, it is well-documented that non-pharmacological cognitive behavioral therapy, relaxation techniques, and good sleeping habits are effective for the treatment of the majority of cases of chronic primary insomnia [52,53]. These cognitive behavioral therapies are considered a gold standard in treating insomnia, and have proved to be as effective as pharmacological therapy, have a maintenance effect after the therapy is ceased, and are better options than pharmacological treatment for chronic insomnia [4]. Nevertheless, pharmacological treatment continues to be utilized by many doctors as the first or only option for treatment [52,53]. This may be attributed to the fact that cognitive behavioral therapy is delivered by a trained health professional, making it a more expensive, laborious and time-consuming therapy [4]. Another important factor affecting therapy preferences is the clinician’s and patient´s knowledge and beliefs; insomnia is often regarded as a symptom rather than a condition itself, and sometimes both clinicians and patients seek to solve the problem as soon as possible, considering pharmacological treatment a time-efficient option, avoiding schedule and time conflicts [54].

The main limitation of the present study is that it was carried out with a small sample size; therefore, although the vast majority of patients in this study had irrational use of medications to treat insomnia, the results cannot be generalized to all clinical contexts of this pathology. Additionally, the study is monocentric and patients seeking medical attention at the SDC may not be representative of the entire population of patients suffering from insomnia. However, this clinic is an important healthcare center with doctors specialized in treating this condition; therefore, the data obtained from patients are important to provide an overview of the clinical practice that is carried out in the country before the patient seeks specialized care.

## 5. Conclusions

This study helps provide an overview of clinical practice with patients with insomnia before they seek specialized care. Based on the present analysis of the use of pharmacological treatments for insomnia by the study participants, 92% had irrationally used their medications to try to relieve their symptoms. Two very common forms of irrational use were found: taking a drug inadequate for the particular patient’s condition (in spite of usually being prescribed by a doctor) and taking a medication longer than the maximum recommended time. The extent of irrational use of medications for insomnia is worrisome, especially because it was greatest with benzodiazepines, which are only recommended for treating severe insomnia in contrast with other safer pharmacological options. Although most participants had acquired their medication by prescription, the drug was inappropriate for treating their condition in most cases. Due to the widespread irrational use of pharmacological treatments by patients with insomnia, public policy should ensure that medical attention is provided to such patients by specialized doctors in primary clinical care. In this way, the risks to health caused by the irrational use of medications can be diminished.

## Figures and Tables

**Figure 1 pharmacy-12-00056-f001:**
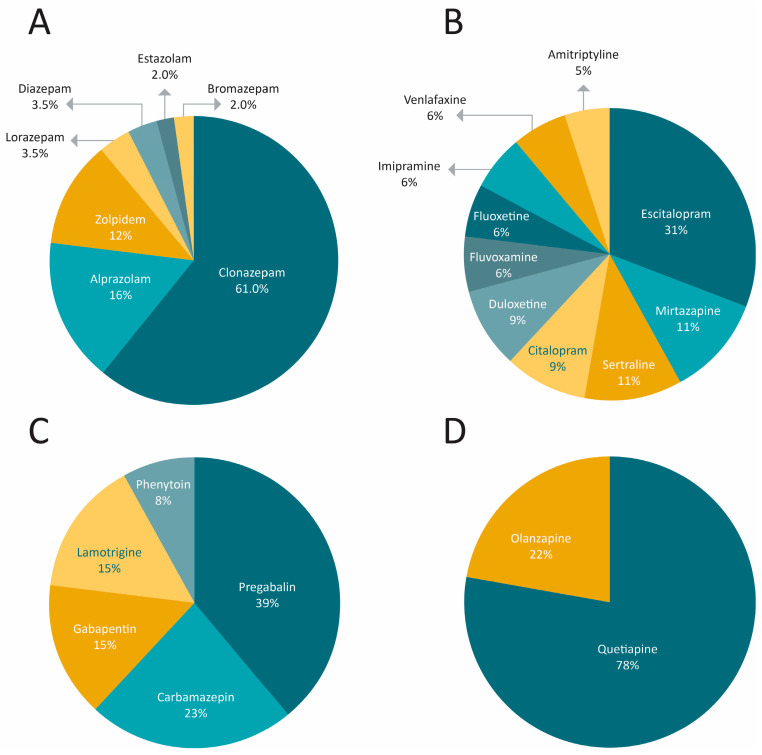
Types and distribution of drugs used by the participants in the study during the 12 months prior to consulting the SDC: (**A**) hypnotics, (**B**) antidepressants, (**C**) anticonvulsants, and (**D**) antipsychotics.

**Figure 2 pharmacy-12-00056-f002:**
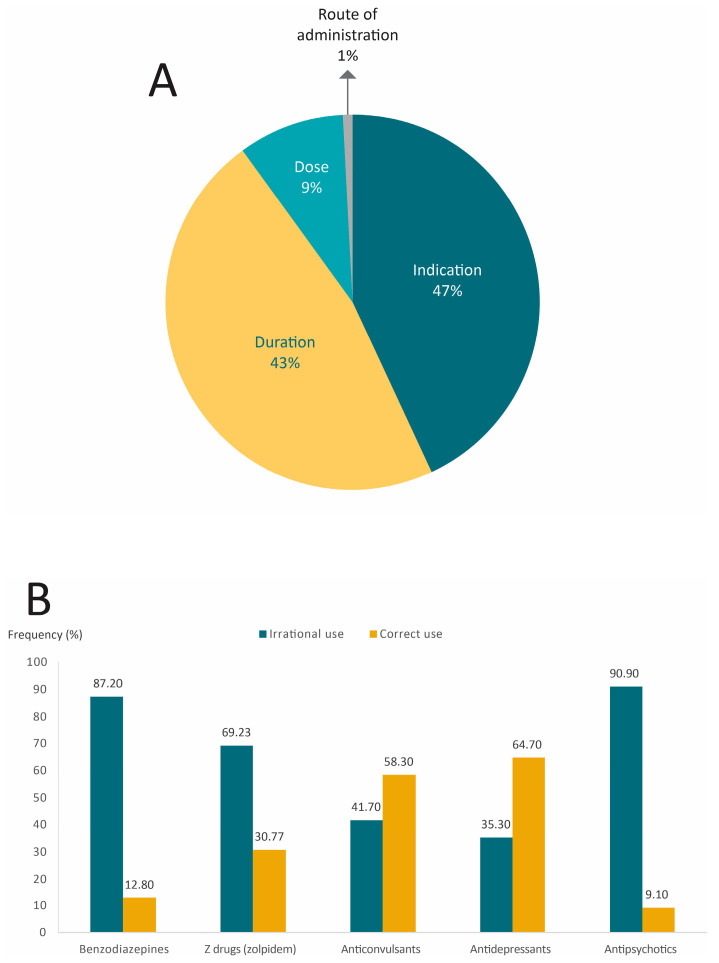
Types and prevalence of irrational use of hypnotics, antidepressants, antipsychotics, and anticonvulsants among individuals with insomnia. (**A**) Type of irrational use, and (**B**) Irrational use by type of drug.

**Figure 3 pharmacy-12-00056-f003:**
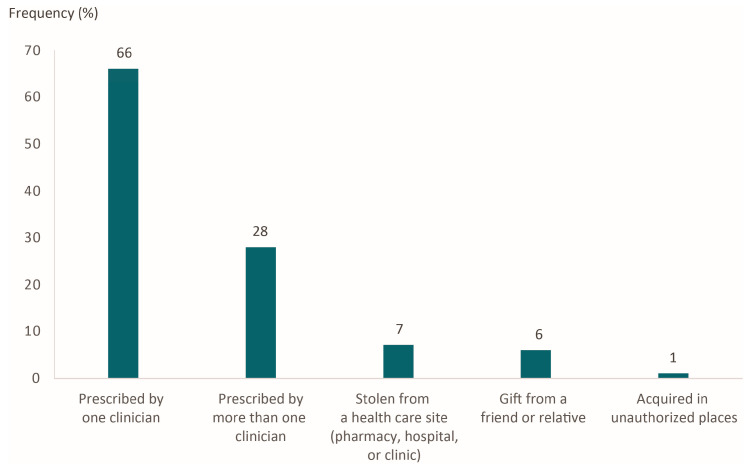
Manner of acquiring medications for the treatment of insomnia.

**Table 1 pharmacy-12-00056-t001:** Sociodemographic characteristics of the participants in the study.

Variable	Frequency% (*n* = 89)
Age (years)	
18–39	12%
40–59	33%
60–79	52%
>80	3%
Sex	
Female	64%
Male	36%
Education	
Illiterate	3%
Elementary School	20%
Junior High School	24%
High School	12%
University	35%
Postgraduate	6%
Occupation	
Home	32%
No work activity	20%
Technical	23%
Professional	11%
Other	14%
Workshift	
No work activity	39%
Morning shift	31%
Afternoon shift	20%
Night shift	2%
Mixed shift	8%
Marital status	
Single	22%
Married	62%
Divorced	8%
Widowed	8%

**Table 2 pharmacy-12-00056-t002:** Associations between medication irrational use and demographic characteristics of patients and the manner of acquiring the drug.

Variable	Misuse (*n* = 82)	Correct Use (7)	OR	95% CI	*p*-Value
Age (years)					
<60	37	3	1.10	0.23–5.21	0.908
>60	45	4			
Sex					
Female	55	3	2.72	0.57–13.00	0.211
Male	27	4			
Level of schooling					
Basic	39	3	Reference		
High School	10	1	1.30	0.12–13.87	0.828
Bachelor’s/Postgraduatedegree	33	3	1.18	0.22–6.25	0.844
Marital status					
With a partner	50	5	0.63	0.11–3.42	0.588
Single	32	2			
Manner of acquiring medications					
Correct	72	6	1.20	0.13–10.39	0.872
Incorrect	10	1			

## Data Availability

Data are contained within the article.

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
