# Peer review of "Irrational Use of Medications among Adults with Insomnia: An Observational Study at a Sleep Clinic in Mexico"

_pharmacy, 2024, doi:10.3390/pharmacy12020056_

Round 1

Reviewer 1 Report

Comments and Suggestions for Authors

The purpose of the article: ‘’ Misuse of medications among adults with insomnia: an observational study at a sleep clinic in Mexico’’, Carvalho-Silva Lucelya et al., is to determine the patterns of misuse of medications among a sample of adult patients with insomnia.  The article highlights widespread medication misuse, particularly benzodiazepines and antipsychotics, among adult patients with chronic insomnia, emphasizing the need for specialized medical attention in primary clinical care to address inappropriate prescriptions and prolonged treatment durations.

Regarding the level of professionalism and quality of English, the article appears well-written and professionally structured. The author effectively covers the issues surrounding medication misuse for insomnia, providing clear references to the literature, and offering practical recommendations for improving patient care. The language is consistent, and the article maintains a logical flow, which contributes to a credible and informative piece of scholarship.

 Title and Abstract - fully reflect its content and research goals.

 Introduction

The introduction of the article provides a comprehensive overview of the prevalence and consequences of insomnia.

Adding related information would significantly supplement this part of the publication - the definition of insomnia, its stages and classification, usual treatment practice: what medications and other types of therapy are used in this case, and the clinical manifestations of the disease itself.

The article mentions "significant long-term health consequences,". Despite the indicated source (early research), it would be more impactful to specify some of these consequences.

 Materials and methods

The methodology described in this study appears thorough and well-structured, with clear details regarding participant selection and analysis.

The statistical analysis section lacks information on specific statistical tests used, making it challenging to assess the appropriateness of the chosen methods.

 Results

The results section is well organized and provides a clear description of procedures and results.

In figures, being transferred from full-color mode, shades of gray are poorly separated, it is necessary to distinguish between shades of gray in the color representation of the figures and increase the scale of the inscriptions in the figures for the best graphical interpretation of the results shown.

The quality of the pictures presented in the effective part is very low - could this be the reason for this - their rough presentation in this version of the publication, with the idea of their subsequent replacement with high-quality pictures when publishing the final version of the article?

 In Table 2, when describing, the author does not indicate the p-value (parameter characterizing the presence of a statistically significant difference or association between the compared parameters). In this case, this table is not informative and necessary in the context of this article.

 Discussion, conclusion

The discussion in this study provides valuable insights into the high frequency of therapeutic drug misuse among insomnia patients, shedding light on the predominant use of benzodiazepines. a few critical points should be considered:

           The study claims that misuse was widespread (92% of participants), but caution should be exercised in generalizing these findings to a broader population. The sample size is relatively small, and the population seeking medical attention at the Sleep Disorders Clinic may not be representative of all individuals with insomnia. – This point should be emphasized when describing the limitations of the study.

 Additionally, the study could benefit from a deeper analysis of the socio-cultural factors influencing medication preferences and explore potential barriers to implementing non-pharmacological treatments.

 In conclusion, it is necessary to emphasize the important, fundamental conclusions that characterize and summarize this study. Thus, part of the information from this section of this article (line 334-344) can be attributed to the discussion.

Author Response

March 15, 2024

Prof. Dr. Jon Schommer

Editor-in-Chief

Pharmacy

Dear Dr. Schommer, we appreciate the comments and observations to our manuscript entitle “Misuse of medications among adults with insomnia: an observational study at a sleep clinic in Mexico”, they help us to improved the quality of the manuscript. Below are the responses to Reviewer 1 comments:

INTRODUCTION

1. Reviewer’s comment: Adding related information would significantly supplement this part of the publication - the definition of insomnia, its stages and classification, usual treatment practice: what medications and other types of therapy are used in this case, and the clinical manifestations of the disease itself.

Response: The information has been added. The new information is shaded in yellow.

2. Reviewer’s comment: The article mentions "significant long-term health consequences,". Despite the indicated source (early research), it would be more impactful to specify some of these consequences.

Response: The comment was attended with a brief explanation of the consequences with different levels of severity. The new information is shaded in yellow.

MATERIALS AND METHODS

3. Reviewer’s comment: The statistical analysis section lacks information on specific statistical tests used, making it challenging to assess the appropriateness of the chosen methods.

Response: We have added the specific statistical test used. The changes are shaded in yellow.

RESULTS

4. Reviewer’s comment: In figures, being transferred from full-color mode, shades of gray are poorly separated, it is necessary to distinguish between shades of gray in the color representation of the figures and increase the scale of the inscriptions in the figures for the best graphical interpretation of the results shown.

Response: Now the figures are in full-color mode and we have increased the scale of the inscriptions.

5. Reviewer’s comment : The quality of the pictures presented in the effective part is very low - could this be the reason for this - their rough presentation in this version of the publication, with the idea of their subsequent replacement with high-quality pictures when publishing the final version of the article?

Response: We have improved the quality of the graphs.

6. Reviewer’s comment: In Table 2, when describing, the author does not indicate the p-value (parameter characterizing the presence of a statistically significant difference or association between the compared parameters). In this case, this table is not informative and necessary in the context of this article.

Response: We have added the p-values in Table 2. We decided to leave the table because these results are alluded to in the discussion.

DISCUSION AND CONCLUSION

7. Reviewer’s comment: The study claims that misuse was widespread (92% of participants), but caution should be exercised in generalizing these findings to a broader population. The sample size is relatively small, and the population seeking medical attention at the Sleep Disorders Clinic may not be representative of all individuals with insomnia. – This point should be emphasized when describing the limitations of the study.

Response: We have added a paragraph with the limitations of the study at the end of the discussion section.

8. Reviewer’s comment: Additionally, the study could benefit from a deeper analysis of the socio-cultural factors influencing medication preferences and explore potential barriers to implementing non-pharmacological treatments.

Response: We have added information in the discussion section to make a deeper analysis of the socio-cultural factors influencing medication.

9. Reviewer’s comment: In conclusion, it is necessary to emphasize the important, fundamental conclusions that characterize and summarize this study. Thus, part of the information from this section of this article (line 334-344) can be attributed to the discussion.

Response: Now, the conclusion highlights and summarizes the important information and results derived from the study.

We appreciate all comments and suggestions.

Regards,

Dr. Juan Manuel Martínez Núñez

Reviewer 2 Report

Comments and Suggestions for Authors

This is a well written manuscript, easy to read, easy to follow statistical analysis and results are clearly reported. Although the  topic of the prescribing of hypnosedatives is not novel or original, this is an interesting look at this very relevant problem and adds to the literature about this problematic area, and once again draws attention to the off licence use of medication, exploiting the pharmacodynamic actions that cause sedation. The conclusions are congruent with the results and referenced literature. The only issue I have is with the word " misuse" as this is usually used to infer misuse by the patient/user, whereas in this paper it's used in a much wider context (which could cause some confusion to the reader). I wonder if  as per line 76, the word irrational ( or inappropriate) could replace the word "misuse"?

Author Response

March 15, 2024

Prof. Dr. Jon Schommer

Editor-in-Chief

Pharmacy

Dear Dr. Schommer, we appreciate the comments and observations to our manuscript entitle “Misuse of medications among adults with insomnia: an observational study at a sleep clinic in Mexico”, they help us to improved the quality of the manuscript. Below are the responses to Reviewer 2 comments:

Reviewer comment: The only issue I have is with the word "misuse" as this is usually used to infer misuse by the patient/user, whereas in this paper it's used in a much wider context (which could cause some confusion to the reader). I wonder if as per line 76, the word irrational (or inappropriate) could replace the word "misuse"?

Response: Thank you for your observation. We have replace the word “misuse” for “Irrational use” that is a more appropriate term. The changes are reflected in the entire document and they are shaded in yellow.

We appreciate all comments and suggestions.

Regards,

Dr. Juan Manuel Martínez Núñez

Reviewer 3 Report

Comments and Suggestions for Authors

The authors present a work regarding the misuse of medicines for the treatment of insomnia. The work is interesting and well-presented. There are some comments that have to do with the results. But generally the results are of interest  

1) Were the patients diagnosed with insomnia and prescribed hypnotics priort o their visit in SDC?

2) Any bias in interpreting the term misuse? 

3) Any drug interactions with other medications?  

4) Did they used any supplements to deal with their condition? 

5) The study sample is relative small. Any limitations of the study's results should be mentioned. 

Comments on the Quality of English Language

n/a

Author Response

March 15, 2024

Prof. Dr. Jon Schommer

Editor-in-Chief

Pharmacy

Dear Dr. Schommer, we appreciate the comments and observations to our manuscript entitle “Misuse of medications among adults with insomnia: an observational study at a sleep clinic in Mexico”, they help us to improved the quality of the manuscript. Below are the responses to Reviewer 3 comments:

1. Reviewer comment: Were the patients diagnosed with insomnia and prescribed hypnotics prior to their visit in SDC?

Response: Yes, the subjects have been diagnosed with insomnia previously and had sought medical attention previously, before they arrived to the SDC. As we described in the methods section, patients were recruited at their first medical visit at the Sleep Disorders Clinic (SDC). The analysis of irrational use was about the medications to treat insomnia (including hypnotic drugs) that the patients had been using during the 12 months prior to admission to the SDC.

2. Reviewer comment: Any bias in interpreting the term misuse?

Response: Other reviewer made the same observation, arguing that “misuse” is usually used to infer misuse by the patient/user, whereas in this paper it's used in a much wider context. The reviewer suggested used the term “irrational use”, which we agree is more appropriate. This change is reflected in the entire document and they are shaded in yellow.

3. Reviewer comment: Any drug interactions with other medications?

Response: We did not evaluate drug interactions with other medications.

4. Reviewer comment: Did they used any supplements to deal with their condition?

Response: We did not explore the use of supplements among the participants.

5. Reviewer comment: The study sample is relative small. Any limitations of the study's results should be mentioned.

Response: We have added a paragraph with the limitations of the study at the end of the discussion section, which addresses the study sample size.

We appreciate all comments and suggestions.

Regards,

Dr. Juan Manuel Martínez Núñez